# *Pseudomonas fluorescens* with Nitrogen-Fixing Function Facilitates Nitrogen Recovery in Reclaimed Coal Mining Soils

**DOI:** 10.3390/microorganisms12010009

**Published:** 2023-12-19

**Authors:** Xin Wu, Xiangying Wang, Huisheng Meng, Jie Zhang, Jamie R. Lead, Jianping Hong

**Affiliations:** 1College of Resources and Environment, Shanxi Agricultural University, Taigu County, Jinzhong 030801, China; wuxinsx@163.com (X.W.); menghuisheng@sxau.edu.cn (H.M.); zhangjie880124@foxmail.com (J.Z.); 2College of Life Sciences, Shanxi Agricultural University, Taigu County, Jinzhong 030801, China; wxyzlb@163.com; 3Center for Environmental Nanoscience and Risk, Department of Environmental Health Sciences, Arnold School of Public Health, University of South Carolina, Columbia, SC 29208, USA; jlead@mailbox.sc.edu

**Keywords:** *Pseudomonas fluorescens*, nitrogen, coal-mining area, soil reclamation

## Abstract

Coal mining has caused significant soil nitrogen loss in mining areas, limiting reclamation and reuse in agriculture. This article studies the effects of organic fertilizer, inorganic fertilizer, and the combined application of *Pseudomonas fluorescens* with the ability of nitrogen fixation on soil nitrogen accumulation and composition in the reclamation area of the Tunlan Coal Mine from 2016 to 2022 under the conditions of equal nitrogen application, providing a scientific basis for microbial fertilization and the rapid increase in nitrogen content in the reclaimed soil of mining areas. The results showed that as the reclamation time increased, the nitrogen content and the composition and structure of the soil treated with fertilization rapidly evolved toward normal farmland soil. The soil nitrogen content increased most rapidly in the presence of added *P. fluorescens* + organic fertilizer (MB). Compared to other treatments (inorganic fertilizer (CF), organic fertilizer (M), and *P. fluorescens* + inorganic fertilizer (CFB)), MB increased total nitrogen (TN) to normal farmland soil levels 1–3 years earlier. The comprehensive scores of MB and CFB on the two principal components increased by 1.58 and 0.79 compared to those of M and CF treatments, respectively. This indicates that the combination of *P. fluorescens* and organic fertilizer improves soil nitrogen accumulation more effectively than the combination of *P. fluorescens* and inorganic fertilizer. In addition, the application of *P. fluorescens* increases the content of unknown nitrogen (UN) in acid-hydrolysable nitrogen (AHN) and decreases the content of amino acid nitrogen (AAN) and ammonia nitrogen (AN). However, there was no significant effect on the content of ammonium nitrogen (NH_4_^+^-N) and nitrate nitrogen (NO_3_^−^-N) in soil-mineralized nitrogen (SMN). When combined with inorganic fertilizer, the contribution of SMN to TN increased by 14.78%, while when combined with organic fertilizer, the contribution of AHN to TN increased by 44.77%. In summary, the use of *P. fluorescens* is beneficial for nitrogen recovery in the reclaimed soil of coal-mining areas. The optimal fertilization method under the experimental conditions is the combination of *P. fluorescens* and organic fertilizer.

## 1. Introduction

As the largest coal producer and consumer in the world, China’s proven reserves are 1.73 trillion tons, accounting for 13.3% of the world’s reserves, as of the end of 2020. It is estimated that the mining output of coal will remain over 3.7 billion tons by 2030, causing an estimated 200 km^2^ of farmland damage per year in China alone [1,2]. At present, land reclamation in coal-mining areas generally involves engineering measures such as stripping, backfilling, digging pads, covering soil, and leveling [3]. After the engineering measures are completed, fertilization and planting are carried out to gradually restore soil productivity. However, a series of engineering measures have everted the soil layers upside down, leading to severe nutrient loss in the topsoil layer, with nitrogen loss exceeding 70% [4,5]. Nitrogen, as an important material structure indicator, is widely regarded by many scholars as an important basis for the success of soil reclamation in coal-mining areas [6,7,8]. Research has shown that the natural recovery of soil nitrogen is slow, and artificial restoration is required for rapid restoration [9]. Among them, reclamation in farmland has the advantages of shorter reclamation time and higher economic benefits compared to reclamation in forest land and grassland, which is currently a research hotspot [10,11].

The supplementation of nitrogen in farmland soil mainly comes from the addition of synthetic fertilizers, which can be inorganic and/or organic [12,13]. Adding nitrogen fertilizers can significantly increase soil nitrogen content, and its rate also increases with application years [14]. However, traditional nitrogen fertilizer application methods have drawbacks such as the loss of soil microbial diversity, changes in the bacterial community composition, and stimulation of soil organic nitrogen mineralization, which may increase nitrogen loss caused by leaching and gas emissions [15,16,17] and can easily have negative impacts on the environment [18,19]. Therefore, there is an urgent need for a method to address the above issues and simultaneously increase the nitrogen content of reclaimed soil. Microbial reclamation technology utilizes the advantages of microbial inoculation to increase nitrogen fixation and available nitrogen content by improving microbial nitrogen fixation activity and accelerating soil organic nitrogen mineralization. It is an effective way to solve a series of problems caused by traditional nitrogen applications. As an important plant rhizosphere growth-promoting bacterium, *P. fluorescens* mainly converts nitrogen in the air into a nitrogen source that can be absorbed and utilized by crops through its own nitrogenase secretion [20,21]. In addition, *P. fluorescens* can also secrete antibiotics such as phenazine-1-carboxylic acid [22], pyoluteorin [23], pyrrolnitrin [24], hydrogen cyanide, and 2,4-diacetylphloroglucinol [25] to inhibit the growth of various pathogenic microorganisms, produce auxin, promote the increase in plant chlorophyll content, and produce forms such as glutathione and ACC deaminase to promote plant growth [26]. *P. fluorescens* has the characteristics of simple nutritional requirements, fast reproduction, and strong competitive colonization ability, and has enormous application potential in agricultural production [27].

At present, the application of inorganic and organic nitrogen fertilizers is usually adopted to quickly increase the soil nitrogen content in coal-mining areas and shorten the reclamation period. Few studies have focused on the use of *P. fluorescens* with nitrogen fixation ability to improve the nitrogen content of reclaimed soil in coal-mining areas. Here, we aim to understand the effect of adding *P. fluorescens* and/or different forms of nitrogen to the soil to accelerate the speed of nitrogen recovery in reclaimed soil.

## 2. Materials and Methods

### 2.1. Study Area

The study area is located in the coal gangue discharge area of the Tunlan Coal Mine, 6 km south of Gujiao City, Shanxi Province, China (Figure 1). It belongs to a warm temperate sub-humid continental monsoon climate. The average annual temperature is 9.50 °C and the average annual precipitation is 475.00 mm. After the completion of coal-mine waste disposal in 2012, soil was taken to cover the coal gangue, with a thickness of approximately 1.00 m, and the soil type is calcareous brown soil, taken from nearby mountains. From 2012 to 2015, soil mixing and plowing were used to ensure the covering soil had similar chemical properties. The chemical properties are as follows: pH of 8.53; organic matter of 3.45 g/kg; total nitrogen of 0.19 g/kg; and available phosphorus of 2.51 mg/kg. From 2016 to 2022, a targeted experiment was conducted to improve the nitrogen content of reclaimed soil through fertilization.

### 2.2. Test Materials

Test crop: maize (Dafeng 30).

Test strains: N64-1 and N137-1 were isolated from farmland soil in Shanxi Province. They were identified as *P. fluorescens* (BioProject ID: PRJNA1048664; PRJNA1048667), with nitrogen fixation amounts of 4.54 and 4.44 mg/L (semi micro Kelvin method) and nitrogenase activity of 41.39 and 40.71 nmol/(mg·h) (acetylene reduction method), and showed good growth trends within the range of 0–40 °C and pH 6–9. There was no antagonism between N64-1 and N137-1. N64-1 and N137-1 were stored in an LB culture medium containing 20% glycerol at −80 °C, respectively.

Test inorganic fertilizers: urea (N, 46%), superphosphate (P_2_O_5_, 16%), potassium chloride (K_2_O, 60%), produced by Shanxi Yefeng Chemical Fertilizer Co., Ltd. (Lvliang City, China).

Test organic fertilizer: decomposed chicken manure (N, 1.45%, P_2_O_5_, 1.14%, K_2_O, 0.85%), produced by Hebei Juntian Biotechnology Co., Ltd. (Shijiazhuang City, China).

### 2.3. Experimental Design

Following the principle of equal nitrogen application (N approximately 18,000 kg/km^2^), four fertilization treatments were set up in the experiment: CF: inorganic fertilizer; CFB: *P. fluorescens* + inorganic fertilizer; M: organic fertilizer; and MB: *P. fluorescens* + organic fertilizer; all four fertilization treatments were planted with maize. Three control treatments were included: IS: no maize planting and no fertilization; CK: planting maize without fertilization; and MS: normal farmland (farmland undisturbed by coal mining), which has been continuously cultivated for more than 30 years. The fertilization amount is approximately 2/3 of the fertilization treatment, and the chemical properties are as follows: pH value of 8.12; organic matter of 17.35 g/kg; total nitrogen of 1.17 g/kg; and available phosphorus of 10.11 mg/kg. CF, CFB, M, MB, CF, and IS treatments are located in the coal reclamation area and the MS treatment is located 1.6 km northeast of the coal reclamation area. Each treatment area is 100 m^2^ (10 m × 10 m), with 3 replicate plots. To avoid fertilization interference, we maintained a distance of 2 m between treatments.

### 2.4. Test Implementation

From 2016 to 2022, the seeds were sown at the end of April and harvested at the end of September. The growth period was 110 ± 3 days, and the seeding density was 6,000,000 plants per square kilometer. The activated N64-1 and N137-1 were inoculated into an LB liquid medium and cultured in a shaking table at 28 °C and 150 rpm for 14–16 h with a viable bacteria number of 10^8^ cfu/mL. The bacteria suspension with *C*_N64-1_: *C*_N137-1_ = 1:1 was prepared by mixing the two in equal volumes. We then applied all fertilizers at one time the day before sowing (Table 1). Maize was weeded once in the jointing period and not irrigated during the growing period.

### 2.5. Sampling and Physicochemical Analysis

During the experiment, soil samples were collected within 24 h after maize harvest, and a total of 147 soil samples were analyzed. Topsoil (0–20 cm) was collected using a five-point sampling method in each plot and then mixed into a single sample. After removing visible plant residues and stones, each sample was divided into two parts: one part was stored at −4 °C for the determination of SMN and the other part was dried and ground for the determination of TN and AHN.

The nitrogen composition of the soil is shown in Figure 2. TN was determined by the Kjeldahl method [28]. NH_4_^+^-N and NO_3_^−^-N in SMN were extracted using 2 mol/L KCl and measured using a flow analyzer (Skalar Analytical, Breda, The Netherlands) [29]. The soil organic nitrogen components were determined using the Bremner organic nitrogen classification method [30], with AHN determined using the Kjeldahl method. THN was obtained by subtracting TN from AHN and SMN. AN was determined using the MgO distillation method, and AN + ASN was determined using the phosphate borax salt buffer distillation method. AAN was determined by ninhydrin oxidation and the phosphate borate buffer distillation method, while ASN and UN were obtained by the subtraction method.

### 2.6. Statistical Analysis

Excel 2007 was used for data processing and SPSS 22 was used for the analysis of variance, data fitting, multiple linear regression analysis, and principal component analysis. Regarding data fitting, with time as the horizontal coordinate and nitrogen content as the vertical coordinate, the r^2^ value was used to determine the fitting type and predict the change in nitrogen content; with SMN and AHN as horizontal coordinates and NO_3_^−^-N, NH_4_^+^-N, and AN, AAN, ASN, and UN as corresponding ordinates, linear fitting was used to show the changes in the relative content of NO_3_^−^-N and NH_4_^+^-N in SMN and the changes in the relative content of AN, AAN, ASN, and UN in AHN. Using multiple linear regression, with TN as the dependent variable and different forms of nitrogen as independent variables (no multicollinearity existed between independent variables), the contribution degree of different forms of nitrogen to soil total nitrogen under cumulative fertilization for 7 consecutive years was explained. Regarding principal component analysis, the number of principal components was determined according to the eigenvalues of the variance contribution rate and the cumulative contribution rate of each component, and the recovery effect of different fertilization treatments on soil nitrogen was evaluated by the comprehensive score of the principal component.

## 3. Results

### 3.1. Changes in TN Content

As shown in Figure 3, with the increase in reclamation time, the TN content of MS and IS treatments varied between 1313.72 and 1387.87 mg/kg and 240.73 and 268.14 mg/kg, with no significant variations. The fitting results showed that due to the lack of external nitrogen input, the TN content of CK treatment decreased year by year at a rate of 4.88 mg/kg/y. The TN content of CF, CFB, M, and MB treatments showed an exponential increase trend, and the difference reached a significant level between 2019 and 2022 (*p <* 0.05). MB treatment is expected to reach the normal farmland soil (MS) level after 10 years, which is 1–3 years earlier than CF, CFB, and M treatments. There was no significant difference in TN content between CFB and CF treatments. Compared with M treatment, the TN content of MB increased by 1.47% to 18.38%, and the difference reached a significant level between 2020 and 2022 (*p <* 0.05).

### 3.2. Changes in SMN Content

As shown in Figure 4, with the increase in reclamation time, the NO_3_^−^-N and NH_4_^+^-N content of MS, IS, and CK treatments showed no significant increase or decrease trend. The fitting results showed that the NH_4_^+^-N and NO_3_^−^-N content showed an exponential increase trend in MB and M treatments, while the NH_4_^+^-N and NO_3_^−^-N contents showed linear and exponential increase trends in CFB and CF treatments, respectively. The order in which different fertilization treatments improve the NH_4_^+^-N and NO_3_^−^-N content in reclaimed soil is MB > M > CFB > CF. In 2022, the NH_4_^+^-N and NO_3_^−^-N content in MB treatment is approximately 75.59% and 47.03% of that in MS treatment. It is estimated that 8 to 10 years after the start of reclamation, MB treatment can reach the level of normal farmland soil (MS). Compared with CF treatment, the NH_4_^+^-N content of CFB increased by 3.77–22.13%, and the difference reached a significant level in 2016 and 2019–2022 (*p <* 0.05). The NO_3_^−^-N content significantly increased by 13.94% and 14.23% in 2021 and 2022 (*p <* 0.05). Compared with M treatment, the NH_4_^+^-N content significantly increased by 16.60% to 33.94% from 2016 to 2022, and the NO_3_^−^-N content significantly increased by 12.79% to 24.63% from 2019 to 2022 (*p <* 0.05).

As shown in Figure 5, the NH_4_^+^-N dominates in the SMN treated with IS and CK treatments, while the NO_3_^−^-N dominates in the SMN treated with MS, CF, CFB, M, and MB treatments, indicating that fertilization significantly affects the relative content of NH_4_^+^-N and NO_3_^−^-N in SMN. The regression equation indicates that the increased SMN content in CFB and CF treatments includes 68.51% and 71.89% NO_3_^−^-N and 31.49% and 42.20% NH_4_^+^-N, while the increased SMN content in MB and M treatments includes 57.34% and 57.80% NO_3_^−^-N and 42.66% and 42.20% NH_4_^+^-N. It can be seen that *P. fluorescens* can reduce the relative content of NO_3_^−^-N in SMN, but the effect is not significant, especially when combined with organic fertilizer, as the relative content of NO_3_^−^-N is only reduced by 0.46%.

### 3.3. Changes in AHN Content

As shown in Figure 6, the AAN and AN content of MS, IS, and CK treatments fluctuated within a small range. MB treatment has the best effect on improving the AAN and AN content of reclaimed soil. The fitting results showed that the AAN content of MB treatment showed an exponential increase trend, while M, CFB, and CF treatments showed a linear increase trend. MB treatment is expected to reach normal farmland soil (MS) levels after 9 years, which is more than 10 years earlier than CF, CFB, and M treatments. The AN content of CF, CFB, M, and MB treatments showed a linear increasing trend. In 2022, the AN content of MB treatment increased by 7.61% to 24.39% compared to CF, CFB, and M treatments. The UN content was the same as the ASN results, and the MS treatment was significantly higher than the fertilization treatment and IS and CK treatments. The ASN and UN content of fertilization treatment did not show an increasing or decreasing trend with the increase in reclamation time. Compared with CF treatment, the AAN content of CFB treatment significantly increased by 9.10% to 17.38%, while the AN and UN content significantly increased by 11.99% and 14.46% only in 2020 (*p <* 0.05). Other time differences did not reach a significant level, while there was no significant difference in ASN content. Compared with M treatment, the content of AAN and AN in MB significantly increased by 6.13–16.77%, 9.48–17.82%, and the content of UN significantly increased by 97.10–117.66% from 2019 to 2022 (*p <* 0.05). The content of ASN only increased significantly by 6.20% and 8.82% in 2020 and 2021 (*p <* 0.05). This indicates that the effect of *P. fluorescens* on AAN content is higher than that of AN, UN is higher than that of ASN, and the effect of the combination of *P. fluorescens* and organic fertilizer on the AHN content in reclaimed coal-mine soils is higher than that of the combination of *P. fluorescens* and inorganic fertilizer.

As shown in Figure 7, the order of soil organic nitrogen components in CK, CF, CFB, M, and MB treatments is AAN > UN > AN > ASN, while in IS and MS treatments, the order is UN > AAN > AN > ASN. It can be seen that fertilization, crop growth, and their interactions are all factors that affect the relative content of organic nitrogen components. As the reclamation time increased, the relative content of AN and AAN in AHN increased in CF and CFB treatments, while the relative content of UN decreased. There was no significant change in the relative content of ASN, compared with CF; the relative content of AAN and AN in the CFB treatment decreased by 33.77% and 10.67%, while the relative content of UN increased by 45.01%. Among the AHN increased by MB treatment, 39.90% is AAN, 53.34% is UN, 5.87% is AN, and 0.59% is ASN. Compared with M treatment, the relative content of UN increased by 35.43%, while the relative content of AAN and AN decreased by 27.26% and 7.86%. The combination of *P. fluorescens* with inorganic and organic fertilizers can reduce the relative content of AAN and AN, increase the relative content of UN, and has no significant effect on the relative content of ASN.

### 3.4. Principal Component Analysis and Regression Analysis of Nitrogen in Reclaimed Soil

To further explore the effect of the application of *P. fluorescens* on nitrogen accumulation and composition in reclaimed soil, principal component analysis and multiple linear regression analysis were conducted on the fertilization treatments.

As shown in Figure 8, based on the eigenvalues > 1, two principal components were extracted, and the variance contribution rates of PC1 and PC2 were 95.20% and 3.00%, respectively, explaining a total of 98.20% variation in the variance variable. The seven treatments were copolymerized into four clusters. In the PC1 axis direction, the clusters were CK, IS treatments, fertilization treatment, and MS treatment, from left to right. The same fertilization treatment is arranged from left to right along the PC1 axis direction in chronological order, indicating that soil nitrogen forms gradually towards normal farmland soil along the time dimension during the reclamation process. The spatial positions of fertilization treatment and MS treatment in the same year, from far to near, are CF, CFB, M, and MB treatments, indicating that *P. fluorescens* is beneficial for the restoration of nitrogen content and composition in the reclaimed soil of coal-mining areas. The combination of *P. fluorescens* and organic fertilizer has the best effect.

As shown in Figure 9, there is a significant difference in the comprehensive score results of fertilization treatment on the two principal components, manifested as MB > M > CFB > CF. The comprehensive scores of MB and CFB treatments increased by 1.58 and 0.79, respectively, compared to M and CF treatments, indicating that the application of *P. fluorescens* can increase the nitrogen content of reclaimed soil. The combination of *P. fluorescens* and organic fertilizer has a better effect than the combination of inorganic fertilizer.

Y(CF) = 0.0081X_1_ + 0.0034X_2_ + 0.0713X_3_ + 0.0061X_4_ + 0.2831X_5_ + 0.1661X_6_ + 0.4620X_7_
(1)

Y(CFB) = 0.0091X_1_ + 0.0041X_2_ + 0.0744X_3_ + 0.0058X_4_ + 0.3124X_5_ + 0.0983X_6_ + 0.4959X_7_
(2)

Y(M) = 0.0112X_1_ + 0.0084X_2_ + 0.0552X_3_ + 0.0065X_4_ + 0.2685X_5_ + 0.1085X_6_ + 0.5416X_7_
(3)

Y(MB) = 0.0106X_1_ + 0.0077X_2_ + 0.0378X_3_ + 0.0068X_4_ + 0.2546X_5_ + 0.3359X_6_ + 0.3466X_7_
(4)

Note: Y:TN; X_1_:NO_3_^−^-N; X_2_:NH_4_^+^-N; X_3_:AN; X_4_:ASN; X_5_:AAN; X_6_:UN; X_7_:THN.

Through the regression equation, it can be seen that the contribution of NO_3_^−^-N, NH_4_^+^-N, and ASN to TN is significantly lower than that of AN, AAN, AUN, and UN. Compared with the CF treatment, the contribution of SMN (NO_3_^−^-N + NH_4_^+^-N) in the CFB treatment increased by 14.78%, while the contribution of AHN (AN + ASN + AAN + UN) decreased by 7.19%. Compared with the M treatment, the contribution of SMN (NO_3_^−^-N + NH_4_^+^-N) in the MB treatment decreased by 7.10%, while the contribution of AHN (AN + ASN + AAN + UN) increased by 44.77%. This indicates that the combination of *P. fluorescens* and inorganic fertilizer will increase the contribution of SMN to TN, and the combination of organic fertilizer will increase the contribution of AHN to TN.

## 4. Discussion

### 4.1. Effect of P. fluorescens on Soil Nitrogen Accumulation

Similar to most research results [31], the TN, SMN, and AHN content of all treatments increased over time. The combination of *P. fluorescens* with inorganic and organic fertilizers did increase the nitrogen content of reclaimed soil in coal-mining areas. This is because *P. fluorescens* N137-1 and N64-1 are autotrophic nitrogen-fixing bacteria that can convert N_2_ from the atmosphere into ammonia and store it in the soil [32], although the nitrogen fixation ability of *P. fluorescens* is not as good as that of symbiotic and combined nitrogen-fixing bacteria such as rhizobia and nitrogen-fixing spirochetes, it has advantages such as low host specificity, wide distribution, and strong adaptability, and they are more conducive to colonization and function in coal-mining reclamation areas with harsh environments. Moreover, *P. fluorescens* can directly reduce the pH value of the surrounding soil by synthesizing organic acids such as indole-3-acetic acid [33], reducing the loss of NH_4_^+^-N caused by acid–base reactions. In the preliminary experiment, we found that *P. fluorescens* is similar to biofertilizers and has the disadvantage of poor stability. Therefore, *P. fluorescens* was applied in combination with inorganic and organic fertilizers to increase its stability in soil [34,35].

The results of this study showed that the growth rate and accumulation of soil nitrogen in the combination of *P. fluorescens* and organic fertilizer were significantly higher than those in the combination of inorganic fertilizer. This is because after inorganic fertilizer (urea) enters the soil, the amide nitrogen is hydrolyzed to NH_4_^+^-N under the action of soil urease. Part of NH_4_^+^-N is directly absorbed by crops and converted into crop tissue nitrogen, while the remaining NH_4_^+^-N is converted into NO_3_^−^-N through nitrification, which is absorbed or leached by crops or directly volatilized [36]. In particular, the ammonia volatilization of urea increased by 9.00–40.70% compared to the slow-release nitrogen fertilizer [37]. Finally, only a small part of amide nitrogen is dissolved in the soil solution, absorbed by the soil through hydrogen bonding, or transformed into soil microbial biomass and organic nitrogen by biological assimilation [38], which remain in the soil.

Therefore, we believe that under the appropriate nitrogen application dose, the increase in soil nitrogen content under CF and CFB treatments mainly comes from the increase in biomass such as roots and leaves caused by the absorption and assimilation of microorganisms and crops [39], rather than the residue of inorganic fertilizers. As an important plant rhizosphere growth-promoting bacterium, *P. fluorescens* can simulate the synthesis of plant hormones to directly regulate plant growth and development. It promotes plant growth by increasing the nitrogen and minerals in the soil that can be utilized by plants. Therefore, the comprehensive score of CFB treatment on two principal components is 0.79 higher than that of CF treatment. Organic fertilizer (chicken manure), on the one hand, controls the mineralization of SON and the fixation of SMN by regulating soil C/N, promoting soil nitrogen to meet crop needs, maximizing soil nitrogen utilization efficiency, and avoiding soil nitrogen leaching and gaseous loss [40,41]. On the other hand, applying organic fertilizer to reclaimed soil with insufficient organic matter content can also have a negative excitation effect, increasing soil nitrogen accumulation. A longer research duration has also shown that under equal fertilization conditions, the soil nitrogen accumulation content of organic fertilizer treatment is much higher than that of inorganic fertilizer treatment [42]. At the same time, organic fertilizer can also provide a more suitable culture medium for soil microorganisms, including *P. fluorescens*, improving soil microbial activity and biomass. This is reflected in the comprehensive score of the MB treatment on the two principal components being 1.58 higher than the M treatment. This research result also confirms the view that organic cultivation is more conducive to microbial nitrogen fixation [43].

### 4.2. Effect of P. fluorescens on Soil Nitrogen Composition

Soil nitrogen can be divided into SMN and SON according to its occurrence form. In terms of soil fertility, NH_4_^+^-N and NO_3_^−^-N are the two most important types of SMN [44], so this study assumes that SMN = NH_4_^+^-N + NO_3_^−^-N. The results showed that the relative content of NH_4_^+^-N and NO_3_^−^-N in MB treatment increased and decreased by 0.46% compared to M treatment, while the relative content of NH_4_^+^-N and NO_3_^−^-N in CFB treatment decreased and increased by 3.38% compared to CF treatment. This indicates that *P. fluorescens* has no significant effect on the relative content of NH_4_^+^-N and NO_3_^−^-N, and this is mainly due to the long-term nitrification and nitrate dissimilation reduction in NH_4_^+^-N and NO_3_^−^-N in the soil, which to some extent achieve dynamic equilibrium [45]. *P. fluorescens* is unable to effectively alter factors such as soil clay mineral types, acidity, and alkalinity, which affect the fixation process of NH_4_^+^-N minerals, and increase or decrease the leaching of NO_3_^−^-N. However, we also found that the contribution of SMN to TN in CF treatment is 0.0115, which is lower than 0.0132 in the CFB treatment, and this is due to the use of *P. fluorescens* in the absence of external organic matter addition, which resulted in a significant portion of soil carbon being consumed as energy, leading to a rapid decrease in soil C/N. After its mineralization exceeded assimilation [46], the supply of NH_4_^+^-N and NO_3_^−^-N significantly increased, thus increasing the contribution of SMN to TN. The contribution of SMN to TN in the MB treatment is 0.0183, which is lower than 0.0196 in the M treatment, which is due to competition between crops and microorganisms in the process of absorbing and assimilating NH_4_^+^-N and NO_3_^−^-N in the soil. The addition of *P. fluorescens* is beneficial to microbial competition. From the perspective of soil nitrogen cycling, microorganisms absorb and assimilate NH_4_^+^-N and NO_3_^−^-N into organic components, thus reducing the contribution of SMN to TN.

At present, people’s understanding of SON is still limited, and there is no method to separate different chemical forms of nitrogen from soil without damaging the components of soil organic nitrogen. This study chose to use the acid hydrolysis method to divide organic nitrogen into THN and AHN for research. The results showed that after 7 years of continuous fertilization, the order of soil organic nitrogen components in CF, CFB, M, and MB treatments was AAN > UN > AN > ASN. Although *P. fluorescens* did not change the order of soil organic nitrogen components, it could increase the relative content of UN and reduce the relative content of AAN and AN. This is because UN is composed of non α-amino acid nitrogen, fatty amines and aromatic amines, nitrogen-containing heterocyclic compounds, and amino acids directly connected to aromatic rings through C-N bonds and its content is not related to the mineralization rate of SMN [47]. As a temporary nitrogen reservoir containing a large amount of easily mineralized organic nitrogen, AN is the main available nitrogen in the soil that can be directly absorbed and utilized by crops in the current season, mainly derived from fixed ammonium, especially newly formed fixed nitrogen [48]. AAN is the main source of available nitrogen absorbed by soil microorganisms and plants, especially small-molecule amino acids that can be directly assimilated and absorbed by microorganisms [49]. AAN and AN are the two most important organic nitrogen components that determine the potential of nitrogen mineralization and are the main sources of mineralization. *P. fluorescens* accelerates the mineralization of AN and AAN, resulting in an increase in the relative content of UN. ASN mainly comes from the cell wall of soil microorganisms, which accounts for approximately 5% of soil organic nitrogen. Although *P. fluorescens* can increase the number of soil microorganisms, its colonization time in reclaimed soil is approximately 80 days. Its remains participate in the internal circulation of soil nitrogen in the subsequent time. After 30 days of sampling and measurement, only a portion continues to remain in the form of ASN in the soil, indicating no significant change in the relative content of ASN. A previous study also indicates that there is no correlation between ASN and other nitrogen forms in the soil [50]. CF and CFB treatments due to the absence of exogenous AHN addition, require microbial mineralization to convert the original AHN in the soil into SMN for crop use after the consumption of SMN. *P. fluorescens* promotes the mineralization of AHN, manifested as the contribution of ANH to TN in CFB treatment being 0.4909, lower than that of CF treatment being 0.5266. Research has shown that the contribution of organic nitrogen components is closely related to the difficulty of mineralization [51,52]. When exogenous AHN is added to M and MB, *P. fluorescens* causes a decrease in the contribution of organic nitrogen components that are easily mineralized by AN and AAN, and an increase in the contribution of organic nitrogen components that are not easily mineralized by UN. Specifically, the contribution of AN + AAN decreases by 0.0313, and the contribution of UN increases by 0.2274.

Previous studies indicate that with the same soil type (calcareous brown soil), the nitrogen content of reclaimed soil in coal-mining areas is only 30.43% of that in cultivated soil in Shanxi Province [53,54]. The natural recovery rate of soil nitrogen is extremely slow [55], and the application of *P. fluorescens* can significantly increase the recovery rate of soil nitrogen. Our study also indicated that *P. fluorescens* can increase the available phosphorus content and diversity of soil bacteria in reclaimed soil [8]. At the same time, the increase in soil nutrients can promote the growth, activity, and diversity of soil carbon-fixing microorganisms in reclamation areas and accelerate the rate of soil evolution from reclamation areas to undamaged soil [56]. It can be seen that *P. fluorescens* has enormous potential for application in soil restoration in coal-mining reclamation areas.

## 5. Conclusions

Comprehensive analysis shows that 7 continuous years of fertilization and cultivation have led to a rapid evolution of nitrogen content and composition structure in reclaimed soil toward normal mature farmland soil. *P. fluorescens* can increase the relative content of UN in soil nitrogen and reduce the relative content of AAN and AN. The combination of *P. fluorescens* and inorganic fertilizer can increase the contribution of SMN in TN, while the combination of *P. fluorescens* and organic fertilizer can increase the contribution of AHN in TN. *P. fluorescens* is beneficial for nitrogen recovery in coal-mining areas, and the optimal fertilization method under this experimental condition is the combination of *P. fluorescens* and organic fertilizer. This study can provide a scientific basis for the restoration of soil nitrogen by *P. fluorescens* in coal-mining areas.

## Figures and Tables

**Figure 1 microorganisms-12-00009-f001:**
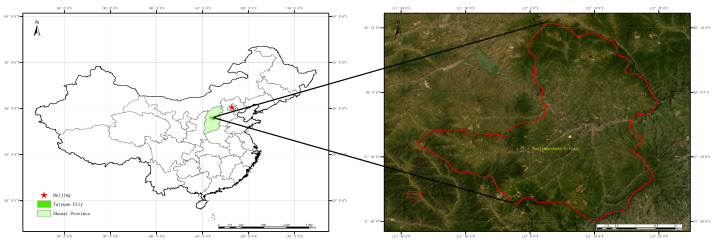
Schematic diagram of geographic location for study area.

**Figure 2 microorganisms-12-00009-f002:**
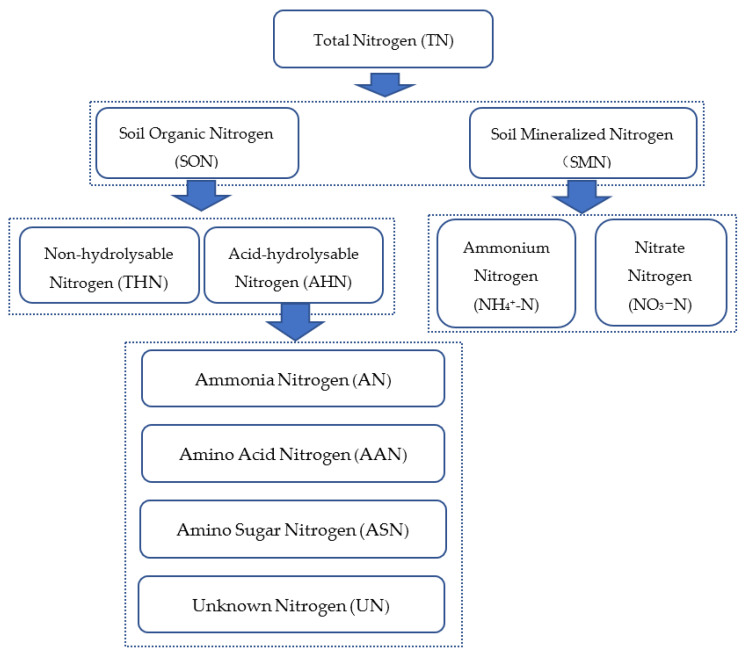
Soil nitrogen composition.

**Figure 3 microorganisms-12-00009-f003:**
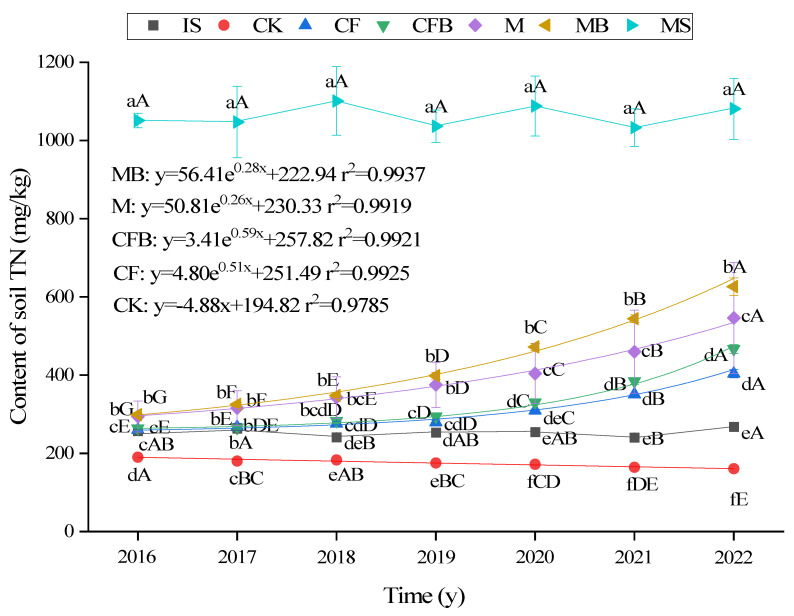
TN content under different fertilization treatments. Note: Lowercase letters indicate significant differences in treatment for the same age group, capital letters indicate significant differences between different years of treatment for the same treatment (*p <* 0.05), and the regression equation is *p <* 0.05, same below.

**Figure 4 microorganisms-12-00009-f004:**
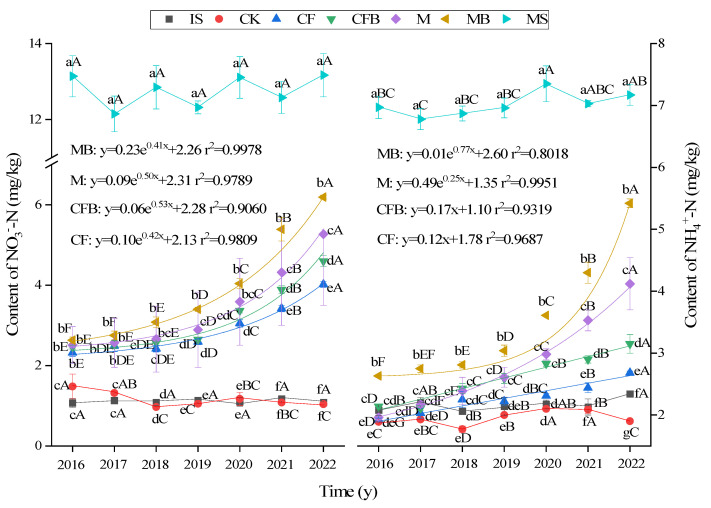
NO_3_^−^-N and NH_4_^+^-N content under different fertilization treatments. Note: Lowercase letters indicate significant differences in treatment for the same age group, capital letters indicate significant differences between different years of treatment for the same treatment (*p* < 0.05), and the regression equation is *p* < 0.05.

**Figure 5 microorganisms-12-00009-f005:**
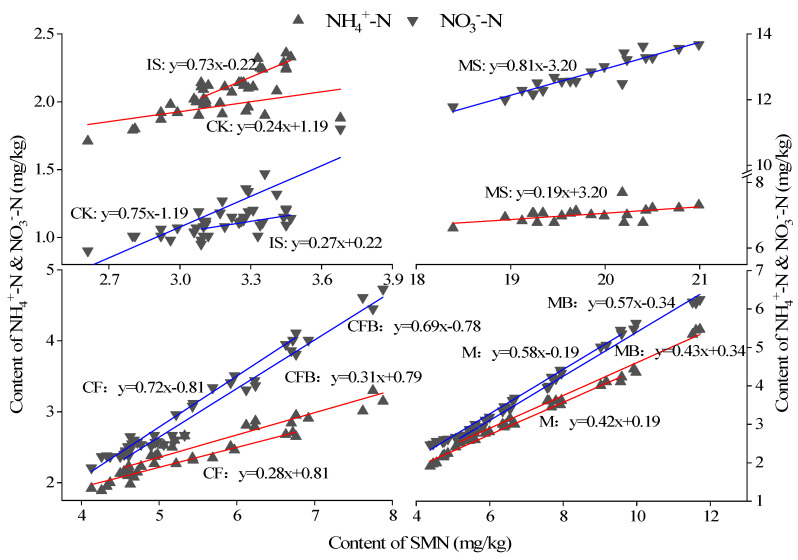
Relative content of NH_4_^+^-N and NO_3_^−^-N in SMN under different fertilization treatments.

**Figure 6 microorganisms-12-00009-f006:**
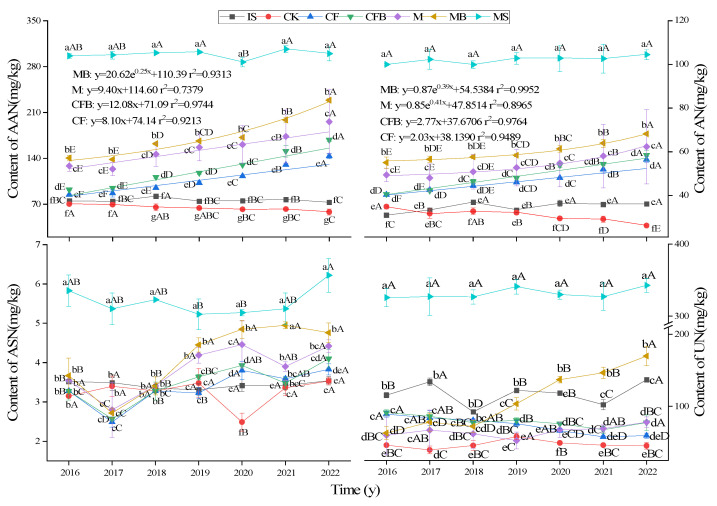
Content of AAN, AN, ASN, and UN under different fertilization treatments. Note: Lowercase letters indicate significant differences in treatment for the same age group, capital letters indicate significant differences between different years of treatment for the same treatment (*p* < 0.05), and the regression equation is *p* < 0.05.

**Figure 7 microorganisms-12-00009-f007:**
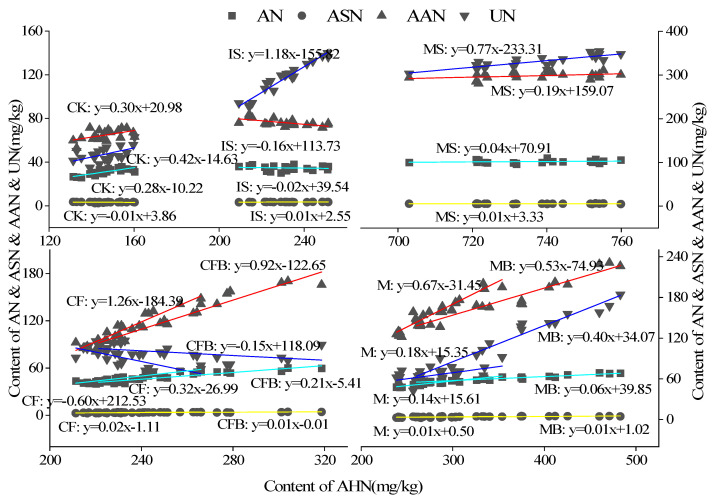
Relative content of AAN, AN, ASN, and UN in AHN under different fertilization treatments.

**Figure 8 microorganisms-12-00009-f008:**
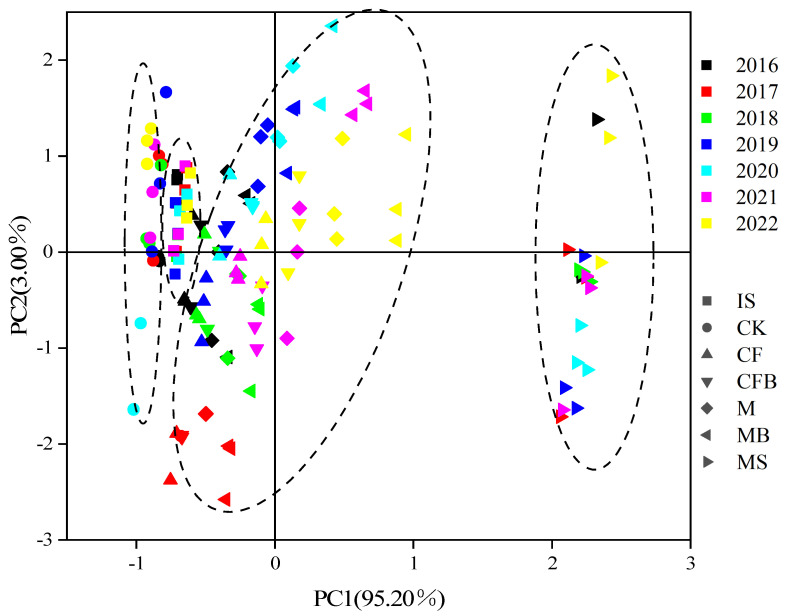
Principal component analysis of reclaimed soil nitrogen.

**Figure 9 microorganisms-12-00009-f009:**
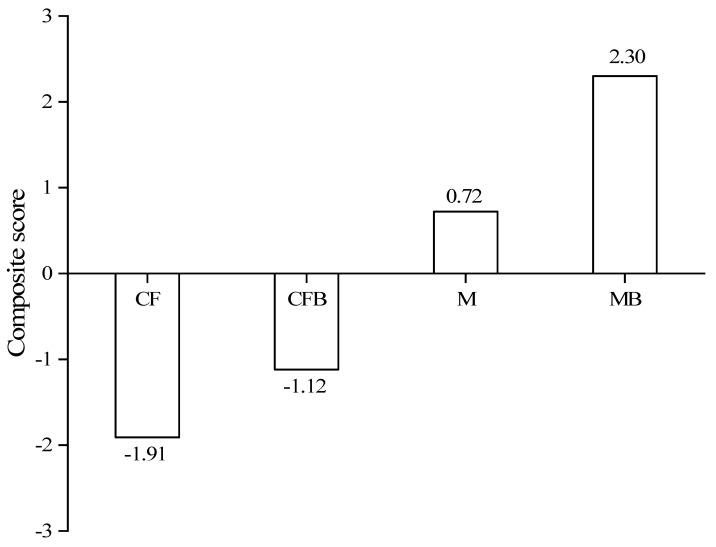
Comprehensive scores of fertilization treatment on two principal components.

**Table 1 microorganisms-12-00009-t001:** Fertilization dose.

Treatments	Urea	Superphosphate	Potassium Chloride	Organic Fertilizer	Bacterial Suspension	LB Culture Medium	Maize
	(kg/km^2^)	(kg/km^2^)	(kg/km^2^)	(kg/km^2^)	(L/km^2^)	(L/km^2^)	
IS							×
CK							√
MS				800,000			√
CF	37,800	85,500	17,000			75,000	√
CFB	37,800	85,500	17,000		75,000		√
M				1,200,000		75,000	√
MB				1,200,000	75,000		√

Note: “×” Indicates not planting maize; “√ “indicates planting maize.

## Data Availability

Data are contained within the article.

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
