# Peer review of "Pseudomonas fluorescens with Nitrogen-Fixing Function Facilitates Nitrogen Recovery in Reclaimed Coal Mining Soils"

_microorganisms, 2023, doi:10.3390/microorganisms12010009_

Round 1

Reviewer 1 Report

Comments and Suggestions for Authors

I have follow suggestions: 

The sequence of used strain have to be deposited in in the NCBI databases.

On page 5, please check the follows: As shown in Figure 2, with the increase of reclamation years, the TN content of MS and IS treatments varied between 1313.72~1387.87 mg/kg and 178.61~194.81 mg/kg, with no significant increase or decrease trend.  IS treatment values is not corect according figure 2.

Author Response

1.I have applied for GenBank accession numbers. Currently under review (submission ID:SUB14017470; SUB14015115), I will add the GenBank accession numbers to the subsequent manuscript as soon as possible.

2.I have corrected that IS treatment values.

Thank you for your help.

Reviewer 2 Report

Comments and Suggestions for Authors

The article discusses various methods of soil reclamation after coal mining in China. The positive effect of introducing the nitrogen-fixing bacterium Pseudomonas fluorescens  has been shown. together with organic nitrogen fertilizer (decomposed chicken manure) to increase the total amount and various forms of nitrogen in the soil. The article is well written, the study is well constructed, based on a large amount of experimental material (over 7 years of analysis) and correct statistical processing.

There are a number of comments to the article, mostly of a technical nature.

1. Page 2, line 3. Available Ref. [1, 3]. Here or earlier, you must indicate the missing Ref. [2].

2. Page 2, line 19. An indication of Ref. is given, without a number.

3. Page 2, lines 28-30. For some reason the names of the substances are written in italics. Among the antibiotics, the species of bacterium Pseudomonas aeruginosa is named.

4. Page 2, lines 30-32. It is not clear why this phrase is given.

5. Page 2, lines 38-40. Indeed, according to the application of Pseudomonas fluorescens there is practically no information in the literature for the reclamation of coal mining soils, but there is extensive literature on other Pseudomonas species. This should be specified.

6. Throughout the text of the article, it is better to write Maize instead of Corn.

7. Section 2.2 contains a description of two Pseudomonas fluorescens lines. It is necessary to clarify how they differed and how they were used in the experiment.

8. Section 2.3 and Table 1: authors are recommended to clarify the dimension kg/hm.

9. Section 2.4. It is necessary to describe the experiment in more detail: indicate the amount of inoculated material (CFE, mL) and the duration of cultivation. Since 2 lines of Pseudomonas fluorescens were isolated and inoculated, it is necessary to clarify how they were introduced - together or separately (if separately, then there should be additional data). In Table 1, the information in the Note is obviously incorrect (mixed up).

10. Fig. 3-6 contain additional numbering before the names of the figures.

11. The Discussion section contains two identical subheadings 4.1.

Author Response

  1. The references in lines 3 and 19 of the second page have been corrected in the manuscript.
  2. The name of the substance in lines 28-30 of the second page has been changed to non-italic, and Pseudomonas aeruginosahas been changed to pyoluteorin for a typo.
  3. Lines 30-32 on page 2 are meant to show that Pseudomonas fluorescenshas been widely used in agricultural production and can be applied with confidence.
  4. In line 38-40 of the second page, the statement about the application of P.fluorescens in coal mining reclamation areas has been deleted and replaced with the advantages of its application in agricultural production, and references are cited to support it.
  5. Corn has been replaced with maize.
  6. Information on N64-1 and N137-1 preservation, culture time, and application methods has been added in sections 2.2 and 2.4.
  7. In Table 1, kg/hm has been changed to the internationally accepted kg/km2, and the information in the Note has been corrected.
  8. The additional numbering included before the title in Figures 3-6 has been corrected.
  9. Section 4.2 has been changed to “Effect of fluorescenson Soil Nitrogen Composition”.

Thank you for your help.

Reviewer 3 Report

Comments and Suggestions for Authors

The manuscript "Pseudomonas fluorescens with nitrogen-fixing function facilitates nitrogen recovery in reclaimed soils of coal mining areas" presents an exploration of microbial intervention in soil recovery. While the manuscript offers insights into the use of Pseudomonas fluorescens for soil nitrogen recovery in coal mining areas, it requires a more focused, detailed, and clear presentation of the methodology, results, and implications before consideration for publication. The current version lacks a clear methodology description and adequate data interpretation, which precludes my recommendation for publication. The authors should substantially revise the manuscript and preferably provide additional supporting data.

Main observations are listed in order of appearance in the text, not importance:

1. The repeated claim that the impacts studied transform the soil in mining areas to "normal farmland soil" is highly speculative. It is unclear what the authors define as 'normal' and what specific parameters are considered. For such a statement, measuring only the parameters provided is insufficient.

2. There are issues in the second paragraph of the introduction. The mention of antibiotics produced by Pseudomonas fluorescens includes “Pseudomonas aeruginosa”. Why are other examples italicized? The source of “synthetic auxin” is unclear; if synthesized by Pseudomonas fluorescens, a reference is needed. The last sentence is inappropriate for an introduction, listing general properties of Pseudomonas fluorescens without specific data or references.

3. The last sentence of the introduction does not align with the study's scope, which is to examine effects, not to improve soil directly.

4. The coordinates provided (37°53′12″N 12°6′13″E) are incorrect, pointing to a location in the Mediterranean Sea near Sicily, Italy.

5. More information is needed about the experimental and control areas, including their proximity and whether experiments were conducted on-site or with removed soil. A satellite image delineating experimental zones would be beneficial.

6. Given the large treatment area (100m2 each repeated 3 times for 7 types of exposure), were geological and physicochemical parameters consistent across all sites?

7. How was bacterial contamination controlled in non-treated areas? Experimental evidence of this control is necessary.

8. Units for Pseudomonas fluorescens and LB culture medium should be specified (kg, l, ml etc.). How were bacteria introduced (freeze-dried preparations or suspensions of bacteria in a nutrient medium, etc.)? Details on the proportion and identification of the two strains (N64-1 and N137-1) are required.

9. Where was the control soil (MS) sourced, and what were its parameters at the experiment's start? The location of this farmland zone relative to experimental plots needs clarification.

10. The preparation of fertilizers, bacteria, and soil nitrogen composition determination must be detailed, including equipment and chemical test specifications.

11. Table 1 has reversed designations in the caption. All figures need clarification on the meaning of letters and error types. The formulas used in graphs and their calculation methods must be explained and discussed in the text.

12. The use of multiple linear regression and principal component analysis requires deeper discussion and clearer interpretation in relation to the study's objectives.

13. The discussion chapters have identical titles and lack in-depth analysis of treatment effectiveness. The real-world applicability of the findings should be elaborated.

Overall, the manuscript is poorly prepared, with numerous grammar, punctuation, and consistency errors. The absence of line numbers hinders specific error referencing. A thorough proofread by a native English speaker is necessary.

Comments on the Quality of English Language

The manuscript, as it stands, suffers from extensive issues related to grammar, punctuation, and consistency. To address these shortcomings, comprehensive proofreading by a native English speaker is essential.

Author Response

  1. The normal farmland soil mentioned in the manuscript refers to the farmland around the experimental area that has not been affected by coal mining activities. The manuscript specifically refers to MS. MS is farmland that has been continuously cultivated for more than 30 years around the coal reclamation area and has the same soil type (calcareous brown soil) . The fertility parameters and geographical location of MS at the beginning of the experiment have been described in chapter 2.3 of the manuscript.
  2. In the second paragraph of the introduction, the antibiotics produced by Pseudomonas fluorescensshould be pyoluteorin, and all antibiotic names have been changed to non-italics. References have been added to support the role of P. fluorescens in promoting crop growth by producing auxin. The advantages of P. fluorescens, such as rapid reproduction, are listed in order to show its great application potential in agricultural production, and corresponding references are attached.
  3. The last sentence of the introduction has been changed to read: The aim is to accelerate the speed of nitrogen recovery in reclaimed soil.
  4. The location of the test area has been corrected and satellite images of the test area have been added. Information such as reclamation mode and soil fertility of the experimental area has been supplemented (2.1Study Area and 2.3 Experimental Design).
  5. After the completion of coal mine waste disposal in 2012, soil was taken to cover the coal gangue, with a thickness of about 1.00 meters, the soil type is calcareous brown soil, taken from nearby mountains.  From 2012 to 2015, soil mixing and ploughing were used to make the soil in the studyarea have similar chemical properties in 2016. To avoid fertilization interference, keep a distance of 2 m between treatments.
  6. The unit of culture medium of P.fluorescens and LB was L. P. fluorescens was introduced as a cell suspension. CN64-1:CN137-1=1:1 in cell suspension. N64-1,N137-1 identification information has been uploaded to the NCBI database and is currently under review (submission ID:SUB14017470; SUB14015115), I will add the GenBank accession numbers to the subsequent manuscript as soon as possible. The above has been recorded in the manuscript.
  7. Organic and inorganic fertilizers are purchased from Shanxi Yefeng Chemical Fertilizer Co., Ltd. and Hebei Juntian Biotechnology Co., Ltd., respectively. Corresponding references have been added to the methods of soil nitrogen determination.
  8. The errors in Table 1 have been corrected.
  9. The formulas and calculation methods used are explained in Section 2.6.
  10. Section 4.2 has been changed to “Effect of fluorescenson Soil Nitrogen Composition”.
  11. By introducing the published results of this research team, it was confirmed that fluorescenscan increase the available phosphorus content, the diversity of soil microorganisms, and the activity of other microorganisms in the reclaimed soil . It shows that P. fluorescens has great potential in soil remediation in coal mining reclamation areas.
  12. The manuscript has been proofread by Professor Jamie R Lead from the University of South Carolina in the United States.

Thank you for your help.

Round 2

Reviewer 3 Report

Comments and Suggestions for Authors

In the revised manuscript, the authors have thoroughly addressed my previous comments. The modifications made have significantly enhanced the clarity of the publication, which will undoubtedly aid readers in understanding its importance in the field. Consequently, I recommend its publication in the journal "Microorganisms".